# Synergistic or Antagonistic Health Effects of Long- and Short-Term Exposure to Ambient NO_2_ and PM_2.5_: A Review

**DOI:** 10.3390/ijerph192114079

**Published:** 2022-10-28

**Authors:** Anna Mainka, Magdalena Żak

**Affiliations:** Department of Air Protection, Silesian University of Technology, 22B Konarskiego St., 44-100 Gliwice, Poland

**Keywords:** air pollutants, NO_2_, PM_2.5_, multi-pollutant approach, mortality, morbidity

## Abstract

Studies on adverse health effects associated with air pollution mostly focus on individual pollutants. However, the air is a complex medium, and thus epidemiological studies face many challenges and limitations in the multipollutant approach. NO_2_ and PM_2.5_ have been selected as both originating from combustion processes and are considered to be the main pollutants associated with traffic; moreover, both elicit oxidative stress responses. An answer to the question of whether synergistic or antagonistic health effects of combined pollutants are demonstrated by pollutants monitored in ambient air is not explicit. Among the analyzed studies, only a few revealed statistical significance. Exposure to a single pollutant (PM_2.5_ or NO_2_) was mostly associated with a small increase in non-accidental mortality (HR:1.01–1.03). PM_2.5_ increase of <10 µg/m^3^ adjusted for NO_2_ as well as NO_2_ adjusted for PM_2.5_ resulted in a slightly lower health risk than a single pollutant. In the case of cardiovascular heart disease, mortality evoked by exposure to PM_2.5_ or NO_2_ adjusted for NO_2_ and PM_2.5_, respectively, revealed an antagonistic effect on health risk compared to the single pollutant. Both short- and long-term exposure to PM_2.5_ or NO_2_ adjusted for NO_2_ and PM_2.5_, respectively, revealed a synergistic effect appearing as higher mortality from respiratory diseases.

## 1. Introduction

Recently, there has been a significant increase in publications on air pollution and human health [1]. As an example, Sun and Zhu [2] in a scoping review analyzed 361 articles covering studies on both short- and long-term health consequences of exposure to various ambient air pollutants. Among ambient air pollutants most commonly investigated are particulate matter (PM) with diameters < 10 μm and <2.5 μm (PM_2.5_, PM_10_), as well as nitrogen dioxide (NO_2_). To a lesser extent, studies included general air pollution of gases such as ozone (O_3_), sulfur dioxide (SO_2_), and carbon monoxide (CO), as well as hazardous air pollutants (HAPs) such as metals and volatile organic compounds (VOC) [3,4,5,6,7,8,9,10,11,12,13,14,15,16]. Health effects are analyzed in a wide range. Dominski et al. [1] ranked the most frequently studied health effects, starting with (1) respiratory diseases including asthma, respiratory infections, respiratory disorders, and chronic obstructive pulmonary disease (COPD). Following this were (2) cardiovascular diseases including hypertension, heart rate variability, heart attack, cardiopulmonary disease, ischemic heart disease, blood coagulation, deep vein thrombosis, and stroke; (3) other diseases such as DNA methylation changes, neuro-behavioral disfunctions, inflammatory disease, skin disease, and disability; (4) pregnancy and children characterized by such parameters as birth weight, infant death, infantile eczema, preterm birth, fertility, pregnancy-induced hypertension, and other diseases; (5) mental disorders: Alzheimer’s disease, Parkinson’s disease, depression and stress, annoyance, autism spectrum disorder (ASD), and reduced cognitive function; (6) all causes, when there are no specific causes of disease, including morbidity, hospital admissions, outpatient visits, emergency room visits, and mortality; (7) chronic diseases: diabetes and chronic respiratory diseases; (8) general health outcomes; and (9) cancer, including bladder cancer, brain tumor, breast cancer, liver cancer, lung cancer, and unspecified cancer.

Thus far, we have learned much about the relationships between air pollution and health. In last year’s updated World Health Organization (WHO) air quality guidelines [17], the limit values were significantly decreased (Table 1). We know that the majority of health effects present a positive correlation to air pollution levels and that the harmful outcomes are evident. The plausibility of these associations is supported by results from experimental exposures to humans, animals, and cells [18]. For some pollutants, the results are ambiguous, there is more than one interpretation, or only limited evidence is available for harmful health effects. Moreover, for some pollutants, especially those characterized by levels below the reference concentration, no health effects are to be expected [19].

There are significant statistical associations between concentrations of individual pollutants and population health outcomes. A single-pollutant approach to air pollution management and research was partly motivated by the Clean Air Act [20], which identifies six criteria air pollutants (CO, tropospheric O_3_, lead—Pb, nitrogen oxides (NO_x_), PM, and SO_2_), that are the motivation of air quality regulations based on monitored air concentrations. WHO air quality guidelines [17] point out that among priorities for policy-relevant scientific questions such as “how, why and for whom do the health effects of air pollution exist?” there has been a call to move from a single-pollutant to a multi-pollutant approach. An amendment from the single-pollutant approach requires a shift in air pollution health research to provide a sound basis for multi-pollutant air quality management.

WHO guidelines [17] point out that studies of multi-pollutant exposures including the examination of additive, synergistic (greater than additive), or antagonistic (less than additive) effects play an important role. The Environmental Protection Agency USA provided definitions for terms that describe various types of toxicologic interactions, including forms of additivity, antagonism, inhibition, masking, potentiation, synergism, and other toxicologic phenomena [18,22], defined below:(1)Additivity is when the “effect” of the combination is estimated by the sum of the exposure levels or the effects of the individual chemicals. The effect may refer to the measured response or the incidence of adversely affected animals. The sum may be a weighted sum (the sum of component doses scaled by their toxic potency relative to the index chemical) or a conditional sum (when the toxic response from the chemical mixture is equal to the sum of independent component responses).(2)Antagonism is when the effect of the combination is less than that suggested by the component’s toxic effects. Antagonism must be defined in the context of the definition of “no interaction”, which is usually dose or response addition.(3)Inhibition is when one substance does not have a toxic effect on a certain organ system, but when added to a toxic chemical, it makes the latter less toxic.(4)Masking is when the compounds produce opposite or functionally competing effects at the same site or sites so that the effects produced by the combination are less than suggested by the component’s toxic effects.(5)Potentiation is when one substance does not have a toxic effect on a certain organ or system, but when added to a toxic chemical, it makes the latter more toxic.(6)Synergism is when the effect of the combination is greater than that suggested by the component’s toxic effects. Synergism must be defined in the context of the definition of “no interaction”, when neither compound by itself produces an effect, and no effect is seen when they are administered together.(7)Confounding is another cause of concern in studies of air pollution. If not adequately addressed, cofounding may either increase the apparel effect of air pollution. Confounding occurs when a confounder, a risk predictor for the outcome of concern, co-varies with the air pollutant being investigated.

A fundamental aspect of the multi-pollutant approach to air pollution epidemiology is that the air pollutant concept is being replaced by a common source perspective. To assess the multi-index analysis of ecological risks from the source perspective, a geographic information system (GIS) and remote sensing (RS) can be used. Among indicators reflecting environmental risks, the pressure from urban expansion, land use, and degradation as well as cropland proportion can be included [23]. One of the sources identified as being responsible for adverse health effects is transport. For automobile traffic, the characteristic pollutants are NO_2_ and PM_2.5_. Both are derived from combustion processes and increase oxidative stress [24]. NO_2_ is often treated as a surrogate for PM [25,26]; on the other hand, 14% to 27% of the measured secondary PM_2.5_ is generated from NO_x_ set of reactions [27,28]. Nevertheless, NO_2_ or PM_2.5_ alone are not sufficient to fully characterize the toxicity of the atmospheric mixture or to fully explain the risk of mortality and morbidity associated with exposure to ambient air pollution [6].

Other factors, besides ambient air quality, that determine the length and quality of life is a healthy lifestyle including both dietary habits [29,30] and physical activity [31].

This paper aims to synthesize worldwide evidence on the health effects of short-term and long-term exposure to both NO_2_ and PM_2.5_ on all-cause and/or cause-specific mortality and morbidity. As most of the studies focus on single pollutants, we aim to present the knowledge gaps in the multi-pollutant approach.

Our intention is to provide an illustrative overview of the available health research performed with respect to the concurrent NO_2_ and PM_2.5_ exposure and should be considered rather in a narrative context, not as a systematic review. To identify publications reporting results from studies on the impact of NO_2_ and PM_2.5_ on health effects, we conducted a broad search of the online databases, particularly with articles published during the last 10 years. Both research articles and literature reviews on the relationship between simultaneous NO_2_ and PM_2.5_ exposure and physical and/or mental health effects were included. Inclusion criteria were (1) cohort studies; (2) long-term exposure metrics, i.e., annual or multi-year averages; (3) mean daily or monthly exposure metrics (short-term exposure); (4) cross-sectional, case–control studies. A good example is the systematic review published by Mills et al. [32], presenting the results of 60 articles that provided estimates of both (1) NO_2_ as a single-pollutant and (2) a two-pollutant model including NO_2_ adjusted for PM and to a lesser extent PM meta-analyses for NO_2_ adjusted. However, within a study [32], PM_10_ was the most used metric (67%). Orellano et al. [10] presented evidence on the physical health effects of short-term exposure to PM_10_ and PM_2.5_, NO_2_, and ozone (O_3_) on all-cause mortality, and PM_10_ and PM_2.5_ on cardiovascular, respiratory, and cerebrovascular mortality. A highly valuable work covering both the physical and mental health effects with respect to NO_x_ exposure available risk ratios was included in the review of Shaw and Heyst [33].

The results presented here do not represent an exhaustive list of the literature on physical and/or mental health, but rather offer an illustration of findings in this area, particularly studies presenting statistic parameters: risk, odds, or hazard ratios. It is worth underlining that risk ratios (RR), odds ratios (OR), and hazard ratios (HR) are three common, but often misused, statistical measures, especially in clinical research, or are misunderstood in their interpretation of a study’s results [34].

Risk ratio (RR) is also known as relative risk. The definition from the Dictionary of Environmental Health [35] states that RR is an expression of the occurrence (such as either a percentage or ratio) of the particular phenomenon in an exposed population compared to the appearance of the same phenomenon in a population that has not been exposed. A RR of 1 means that the probability or risk of an event occurring in either population when compared to the other is even, and therefore both groups have the same amount of risk. If the risk is doubled, the RR is 2, and if it is halved, it is 0.5.

Odds ratio (OR) is a statistical technique used in epidemiological case–control studies derived by dividing the odds of an event happening in the exposed group by the odds of the same event happening in the control group [35]. OR provides a measure of the strength of association between two variables on a scale, with 1 being no association, above 1 being a positive association, and below 1 being a negative association. While risk reports the number of events of interest in relation to the total number of trials, odds report the number of events of interest in relation to the number of events not of interest. Stated differently, it reports the number of events to nonevents. For example, the risk of flipping a coin to be heads is 1:2 or 50%; the odds of flipping a coin to be heads is 1:1, as there is one desired outcome (event), and one undesired outcome (nonevent). Misreporting of the OR as the RR, then, can often exaggerate data. It is important to remember that OR is a relative measure just as RR, and thus sometimes a large OR can correspond with a small difference between odds [36].

Hazard ratio (HR) is a similar but distinct measure. It concerns rates of change as a comparison of two hazards. It can present how quickly two survivorship curves diverge through comparison of the slopes of the curves. An HR of 1 indicates no divergence—within both curves, the likelihood of the event was equally likely at any given time. An HR not equal to 1 indicates that two events are not occurring at an equal rate, and the risk of an individual in one group is different than the risk of an individual in another at any given time interval. An important aspect of HRs is the proportional hazard assumption. To report a singular hazard ratio, it must be assumed that the two hazard rates are constant [36]. Regardless of the value of RR/OR/HR, the interpretation should only be made after determining whether the result provides statistically significant evidence towards a conclusion (verified by a *p*-value < 0.05 or 95% confidence interval). Keeping these principles and the RR/OR/HR framework in mind minimizes bias and prevents erroneous conclusions being drawn from the results of a published study on different samples. Table 2 shows the characteristics of RR/OR/HR ratios along with correct and incorrect use.

## 2. Considerations and Discussion

Air pollution has an impact on various health effects. The simplest division includes mortality and morbidity. Mortality reflects the reduction of life expectancy, while morbidity relates to illness occurrence. Mortality is the most studied health endpoint in association with air pollution [37]. The analysis includes all-cause and cause-specific mortality. Other terms used for this indicator are premature death, additional mortality, and death postponed. In all of these metrics, the health effect is expressed by the number of deaths [38]. Morbidity indicator estimates changes in new or existing diseases in a target population [39]. Low air pollution levels can impact on the health of susceptible and sensitive groups, especially with already fragile immune systems (e.g., immunodeficiency, asthma, malnutrition, old age, infancy) [40].

Studies showed a relationship between air pollutants and adverse health effects that appear after short-term (acute) or long-term (chronic) exposure [41]. Short-term exposure to air pollutants is closely related to COPD, cough, shortness of breath, wheezing, asthma, respiratory disease, and high rates of hospitalization (a measurement of morbidity). The long-term effects connected with exposure to air pollution are chronic asthma, pulmonary insufficiency, cardiovascular diseases, and cardiovascular mortality. According to a Swedish cohort study [42], diabetes seems to be induced after long-term air pollution exposure. Moreover, air pollution appears to have various malign health effects in early human life, such as respiratory, cardiovascular, mental, and perinatal disorders, leading to infant mortality or chronic disease in adult age [43]. Exposure to pollutants in utero increases the risk of neuro-developmental delay. Childhood exposure has been inversely associated with neuro-developmental outcomes in younger children, as well as with academic achievement and neurocognitive performance in older children. In older adults, air pollution is associated with accelerated cognitive decline [44].

There is also emerging evidence that air pollutants may adversely affect cognitive development [45], cognitive performance [15,44,45,46,47], and stress [48] and psychological well-being [49,50]. Air pollution impairs verbal tests, and the effect becomes stronger as people age. Cognitive decline or impairment are risk factors for Alzheimer’s disease and other forms of dementia, especially for elderly persons [47,51]. The wide-ranging impacts of air pollutants on brain health and functioning may support the hypothesis of an association with clinically relevant mental health outcomes [9].

Acute and long-term effects are partly interrelated; however, long-term effects are not the sum of short-term effects [26]. The effects of long-term exposure are much greater than those observed for short-term exposure, suggesting that effects are not just due to exacerbations, but maybe also could be due to the progression of underlying diseases. On the other hand, the effects of continued short-term exposure may contribute to the initiation or exacerbation of the chronic disease, and those affected by the acute exposures may reflect a distinct, susceptible subgroup with underlying or existing disease or unrecognized vulnerability [26].

### 2.1. Health Effects of PM_2.5_

PM is a widespread air pollutant, consisting of a mixture of solid and liquid particles suspended in the air, covering a wide range of sizes and chemical compositions [52]. PM is considered so hazardous that in 2016, the WHO and the International Agency for Research on Cancer (IARC) classified ambient PM as Group 1—compounds carcinogenic to humans. The authors of the IARC monograph [53] underline that PM exposure from different sources features mutagenic and carcinogenic effects in people. Unfortunately, the number of air monitoring ground stations is limited, and the spatial distribution is discontinuous, and thus to obtain a fine-grained spatiotemporal distribution of PM_2.5_, a retrieval model can be used [54]. In an estimation of future global mortality from changes in air pollution, Silva et al. [55] predicted 55,600 (−34,300 to 164,000) deaths in 2030 and 215,000 (−76,100 to 595,000) in 2100 due to PM_2.5_ worldwide (countering by 16% the global decrease in PM_2.5_-related mortality.

Fine particles with a diameter < 2.5 μm (PM_2.5_) are considered one of the leading environmental health risk factors due to the potential penetration of particles deeper into the lungs [56]. Total lung deposition of PM_2.5_ particles is about 60% for ultrafine particles and 20% for fine particles [57]. In adults at rest, the nasal deposition of PM_2.5_ is about 20% and increases to 30–40% during exercise. Lower values (about 10–20%) are reached in children aged 5–15 years [58]. Once deposited in the lung, most particles are removed through several clearance mechanisms. Insoluble particles deposited on ciliated airways are generally removed from the respiratory tract by mucociliary activity within 24–48 h [57]. Several PM_2.5_ can also be absorbed into the bloodstream through alveolar capillaries, causing lung and systemic inflammation [59,60,61]. The inflammatory reactions of the lungs and bronchi are due to oxidative stress produced by increased levels of reactive oxygen species (ROS) that responsible for many of the cardiovascular and respiratory health effects [62]. Among the mechanisms related to cardiovascular outcomes, the most influential is the release of pro-oxidative and proinflammatory mediators from the lungs into the circulation and autonomic nervous system imbalance [63].

#### 2.1.1. Short-Term Exposure to PM_2.5_

Short-term exposure to PM_2.5_ evidences a positive relationship with hospital admissions or emergency room visits for cardiovascular outcomes or respiratory effects, increasing from a 0.5 to 3.4% per 10 μg/m^3^ rise in PM_2.5_ levels [37]. Significant associations between short-term (averaging time over 2 or 24 h) PM_2.5_ concentrations and myocardial infarction [27,64] cardiac arrhythmias as well as ischemic stroke [65] were observed. Studies on specific cardiovascular diseases indicate that ischemic heart disease, and congestive heart failure may be influential for the observed associations. Although estimates from studies of cerebrovascular diseases are less precise and consistent, ischemic diseases are more strongly associated with PM_2.5_ compared to hemorrhagic stroke. The available evidence suggests that cerebrovascular effects occur at short lags (approximately 1 day), while effects at longer legs are rarely evaluated. Cardiovascular hospital admissions are especially reported in areas with concentrations ranging from 7 to 18 μg/m^3^ [22,27]. Toxicological studies associated with PM_2.5_ exposure show reduced myocardial blood flow during ischemia and altered vascular reactivity, providing myocardial ischemia. In addition to ischemia, plausible biological mechanisms include increased right ventricular pressure and decreased myocardial contractility in the association between PM_2.5_ and congestive heart failure. Additionally, systemic inflammation and oxidative stress are cited [55,66,67].

In a meta-analysis of 33 studies from China addressing the short-term effects of air pollution, a 10 µg/m^3^ increase in PM_2.5_ was associated with a 0.38% increase in total mortality [68]. Han et al. [69], on the basis of the measurements conducted from 2015 to 2019 in 296 cities across China, estimated that long-term exposure to PM_2.5_ levels exceeding current WHO guidelines (5 µg/m^3^) was associated with 17% average all-cause mortality. Data for China are relevant to analysis performed, for example, in Poland, as PM concentrations in China are far higher than the Western European or North American averages but comparable to results from Poland [68]. The estimated short-term effects of daily PM_2.5_ concentrations on mortality were even more profound than in Western Europe or China. In the short-term exposure to PM_2.5_, the number of emergency hospitalizations for pneumonia increased by 1.7% per 10 µg/m^3^ of PM_2.5_ [70]. In contrast, the increase in the number of deaths for Tricity and Warsaw (Poland) were 2.1% and 2.6% per 10 µg/m^3^ PM_2.5_, respectively [71]. Several lines of evidence suggest that PM_2.5_ promotes and exacerbates allergic disease, which often underlies asthma [72]. A burst of reactive oxygen species induced by PM_2.5_ was found in the neutrophils of asthmatic patients [73]. Braithwaite et al. [9] in a systematic review included 11 studies on short-term (<6 months) associations between eligible mental health outcomes and PM_2.5_.

#### 2.1.2. Long-Term Exposure to PM_2.5_

Longer-term effects of PM_2.5_ exposure could be greater than the immediate ones [71]. PM2.5 is recognized as a key air pollutant significantly related to premature mortality attributed to cardiovascular diseases and lung cancer [74,75]. The results of the European Study of Cohorts for Air Pollution Effects (ESCAPE) showed higher overall mortality due to long-term PM_2.5_ exposure, with statistically significant associations also reported for individuals exposed to PM_2.5_ concentrations below the European threshold of 25 µg/m^3^ [76]. Data from many European cohorts found an association between long-term exposure to PM_2.5_ and lung and kidney cancer [77,78,79,80,81,82,83]. At concentrations exceeding the threshold value, pediatric PM_2.5_ exposures deliver health interventions before the development of obesity and identify and mitigate environmental factors influencing obesity and Alzheimer’s disease. Braithwaite et al. [9], through the analysis of nine articles, support the hypothesis of an association between long-term (≥6 months) PM_2.5_ exposure and depression, as well as anxiety. Calderón-Garcidueñas et al. [84] pointed to diffuse neuroinflammation, damage to the neurovascular unit, and the production of autoantibodies to neural and tight-junction proteins as the worrisome findings in children chronically exposed to concentrations above PM_2.5_ current standards, potentially constituting significant risk factors for the development of Alzheimer’s disease later in life. The results of the REVIHAAP (Review of Evidence on Health Aspects of Air Pollution) project [26] point to additional systemic health effects beyond the respiratory and cardiovascular systems—for example, effects on the central nervous system, the progression of Alzheimer’s and Parkinson’s diseases, developmental outcomes in children, and reproductive health outcomes such as low birth weight.

#### 2.1.3. PM_2.5_ Health Effect Mechanism

There are different mechanisms of effect considering both short- and long-term exposure. The health effects of PM_2.5_, as well as the mechanisms underlying these effects, were investigated by Feng et al. [85] in 132 articles published from 2005 to 2015. The central mechanisms of harmful effects of PM_2.5_ are oxidative stress (intracellular), mutagenicity/genotoxicity, and inflammation [85]. Initially, across the absorption of PM_2.5_ by the targeting cells, toxic effects ensue due to the release of organic chemicals from the pollutant, which is metabolically activated by enzyme systems. Oxidative stress through free radicals and activation of inflammatory cells is an important mechanism [86]. Furthermore, impairment of the antioxidant system is an adverse effect and could be a mechanism for damage. Some organic extracts from PM_2.5_, such as polycyclic aromatic hydrocarbons (PAHs), are responsible for mutagenicity, including DNA damage responses, promoting changes in the biochemistry and physiology of cells. However, one of the main mechanisms of almost all the adverse health effects of PM_2.5_ is inflammation. Systemic inflammation seen through bio-markers (e.g., C-reactive protein) is a consequence induced by PM_2.5_. Moreover, several studies have shown a decrease in immune function caused by PM_2.5_ [85]. The modifications in physiological and biochemical functions, as well as damage to some tissues and organs, are factors that can lead to the development of several negative consequences, including cardiovascular diseases [85].

### 2.2. Health Effects of NO_2_

NO_2_ has low water solubility (0.037 cm^3^/cm^3^ H_2_O at 35 °C), and therefore a large fraction of inhaled NO_2_ could be deposited in the peripheral airways. Inhaled NO_2_, or a component or reaction product of NO_2_, could subsequently be delivered via tissue absorption and transfer across the blood–gas interface to the blood; therefore, systemic effects are possible [87]. It is absorbed along the entire respiratory tract, but exposure studies indicate that the major target site for the action of NO_2_ is the terminal bronchioles. Despite laboratory, clinical, and epidemiological research, the health effects of NO_2_ exposure on humans are not well understood. The toxicological evidence suggests that increased susceptibility to infection; functional deficits from effects on airways; and deterioration of the status of persons with chronic respiratory conditions, including asthmatics, are of potential concern [88].

#### 2.2.1. Short-Term Exposure to NO_2_

In the short-term exposure, a 0.3% increase in total daily mortality was reported per 10 µg/m^3^ NO_2_ increase [88]. The strongest short-term effects of NO_2_ exposition are respiratory hospital admissions (all ages), and the following are all-cause mortality (all ages). Adebayo-Ojo et al. [4] point to a significant increase in hospital admissions for respiratory disease per lower increase in NO_2_ concentration (interquartile range IQR of 7.3 µg/m^3^). A positive association among all ages was 2.3; however, among children under 15 years old, the estimate increased to 3.1%. Cardiovascular hospital admissions are also included; however, a sensitivity analysis showed that in the case of cardiovascular effects, PM is more indicative. The study in Poland [71] showed that the acute health outcomes associated with NO_2_ exposure might be worse than anticipated [76].

Tsai and Yang [89] found a significant association between hospital admissions for pneumonia and PM_2.5_ in Taipei, Taiwan. Polish findings highlight the prominent impact of NO_2_ on pneumonia-related hospitalizations. Pneumonia is an important cause of death, especially in the sensitive groups of young children and the elderly. In 2017, worldwide, 15% of all deaths of children under 5 were caused by pneumonia [90]. In patients with asthma, NO_2_ potentiates bronchial responsiveness [91] and triggers both allergen-dependent [92] and -independent [93] eosinophilic inflammation. At concentrations ≥ 2 ppm, NO_2_ disrupts the tracheobronchial epithelial monolayer [94] and modifies the severity of viral infections [95]. In the Polish study [71], the greatest immediate effect on pneumonia-related hospitalizations in all investigated agglomerations had NO_2_, and a NO_2_ increase of 10 µg/m^3^ was associated with up to an 11% increase in the number of daily pneumonia-related hospitalizations.

#### 2.2.2. Long-Term Exposure to NO_2_

In the ESCAPE project [96], the long-term effect of NO_2_ on natural-cause mortality was not shown. On the other hand, Faustini et al. [96], in a meta-analysis of 19 studies, demonstrated that an annual increase in NO_2_ of 10 µg/m^3^ was associated with a 4.1% increase in natural mortality. The magnitude of NO_2_ effects on mortality was similar to that of PM_2.5_ [96]. Two large cohort studies [97,98] examined the relationship between lung function growth in children and long-term exposure to NO_2_. Although in California [98] the mean NO_2_ concentrations were rather low (from 7.5 to 71.4 μg/m^3^), and in Mexico City [97] were rather high (51–80 μg/m^3^), both demonstrated deficits in lung function growth in children associated with NO_2_ exposition. In Norway and Sweden, exposure to NO_2_ from traffic was associated with decreased lung function in children 9–10 years old in Oslo [99] and 8 years old in Stockholm [100].

#### 2.2.3. NO_2_ Health Effect Mechanism

The main mechanism of NO_2_ toxicity has been suggested to involve lipid peroxidation in cell membranes and various actions of free radicals on structural and functional molecules. The concentrations over 0.2 ppm produce the above-mentioned adverse effects in humans, while concentrations higher than 2.0 ppm affect the cytotoxic T lymphocytes (CD8+ T cells) and natural killer cells (NK cells) that produce our immune response [41].

Associations between NO_2_ pollution and mortality [71,101], as well as the exacerbation of asthma [102], enhance the allergic response to inhaled allergens [103,104], manifesting as respiratory viral infection and its associated inflammation. Spannhake et al. [95] investigated the interactive effects of human rhinovirus type 16 (RV16) and the NO_2_ on markers of proinflammatory activity in human bronchial and nasal epithelial cells. Dose–response experiments for NO_2_ indicated that 3 h exposure to concentrations from 1.0 to 3.0 ppm induced cytokine release from bronchial epithelial cells in a dose-dependent manner with a minimal effect on cell viability. Additionally, the authors of [95] analyzed the interaction between asthmatic symptoms and viral infection within their common epithelial cell targets in the upper and lower respiratory tracks. The results demonstrated that the expression and release of IL-8 (interleukin 8, a pro-inflammatory cytokine that has a role in neutrophil activation and has been identified within the pathogenesis and progression of the disease [105]) in response to combined infection and NO_2_ exposure was significantly higher than the sum of release from the cells that underwent only infection or only NO_2_ exposure.

NO_2_ damages the lung and proteins vital to its function. The mechanism includes exposure of α-1-proteinase inhibitor (α1PI) to NO_2_, which resulted in a 50% loss of immunoreactivity with either monoclonal or polyclonal antibodies in an enzyme-linked immunosorbent assay at molar ratios of NO:α12PI of 100:1 and greater. Additionally, the results of parallel O-phthalaldehyde and bicinchoninic acid protein assays as well as amino acid analysis on control and NO_2_-exposed α1PI suggested a reactivity of NO_2_ with lysine residues [106].

### 2.3. PM_2.5_ and NO_2_

The epidemiological evidence has consistently shown that the NO_2_ associations do not reflect adverse effects of NO_2_ itself but rather the health effects of other air pollutants, mainly PM or other components of the complex mixture of traffic-related air pollutants [32]. Predominantly, this is due to the strong correlations between NO_2_ and other combustion-derived air pollutants, especially PM. Many studies [7,14,26,107,108,109,110,111,112,113,114] underline a positive correlation between NO_2_ and PM_2.5_. In some studies, NO_2_ has been proposed as a surrogate for PM [115,116]; in others, the confounding effects of PM_2.5_ are also underlined [88,117]. Based on 7 years of air pollutant measurements in Iran, analyzed by the AirQ+ model, Naghan et al. [101] observed that the short-term effects of PM_2.5_ on health were greater than those of NO_2_. However, the long-term effects of NO_2_ were greater than PM_2.5_.

A number of studies confirm the relationship between multiple pollutants. For example, in 17 Chinese cities (characterized by average NO_2_ levels of 26–67 µg/m^3^), each short-term 10 µg/m^3^ increase in NO_2_ corresponded to a 1.63% increase in mortality [118]. The association stayed significant when adjusted for PM [118]. In comparison with Chinese findings [118], research performed in Poland [71] showed an increase in daily mortality of 1.7% in Cracow and of 3.9% in the Tricity per 10 µg/m^3^ NO_2_ increase; however, the effect of NO_2_ was independent of other air pollutants including PM_2.5_. In a Dutch birth cohort, NO_2_ and PM_2.5_ were highly correlated (0.93), and a regression model confirmed the association with some outcomes of asthma and allergy during the first 4 years of life [119]. Interestingly, NO_2_ is likely a potential modifier for the association between PM_2.5_ and the risk of inhaled allergies because the risk estimates for PM_2.5_ in higher NO_2_ (≥42.0, μg/m^3^) are statistically different from those in lower NO_2_ levels (*p* = 0.046) [120]. However only a few articles have investigated the relationship between PM_2.5_ and NO_2_, and our analysis identified only eight publications with specific data on RR, OR, or HR (Table 3). In our review, only three articles [16,114,121] covered concentration ranges exceeding EU guidelines (Table 1) for both pollutants.

The health outcomes have been grouped among non-accidental mortality [10,16,76,110,122,123], cardiovascular disease mortality [10,16,114,123], respiratory disease mortality [10,16,123], and additionally stroke [123] and chronic kidney diseases [121]. Table 4 and Table 5 present the comparison of RR, OR, and HR between NO_2_ (Table 4) and PM_2.5_ (Table 5) adjusted for PM_2.5_ and NO_2_, respectively. Generally, the highest risk of mortality is connected with congenital heart diseases (cardiovascular mortality) and pneumonia (pneumonia), both reported in [16]. Wang et al. [16] analyzed the associations between ambient air pollution and the number of deaths among children under 5 years old. The authors found that the associations between pollutants and infant deaths in Beijing (between January 2014 and September 2014) were more pronounced than those in children aged 1–5 years at lag1–lag2 and lag01–lag02. In the case of NO_2_ in the multi-pollutant analysis, the OR was higher at all-cause mortality and pneumonia-related mortality than for a single pollutant. However, in the case of mortality caused by congenital heart diseases, the adjustment of PM_2.5_ brought an antagonistic effect in comparison to NO_2_ alone. An antagonistic relationship was also found for PM_2.5_ exposure adjusted for NO_2_, for all-cause mortality, congenital heart diseases, and pneumonia-related mortality; however, PM2.5 exposure was not statistically significant. As presented in Table 3, Table 4 and Table 5, for congenital heart disease, OR were 1.267 (95% CI 0.643–2.404) for PM_2.5_ adjusted for NO_2_ and 1.667 (95% CI 1.011–2.738) for NO_2_ adjusted for PM_2.5_, while for single pollutants were 1.155 (95% CI 0.953–1.390) and 1.383 (95% CI 1.113–1.718) for PM_2.5_ and NO_2_, respectively. Since the confidence interval (CI) included the value 1, we cannot be sure that the influence of the PM_2.5_ increase was statistically significant. Statistically significant positive associations were observed for NO_2_. Moreover, the pneumonia-related associations of NO_2_ were stronger than those observed for overall death.

Jerret et al. [123] in California cohort studies including 73,711 subjects used land use regression (LUR) models to evaluate the association between air pollution, including PM_2.5_ and NO_2_ and several causes of death, such as cardiovascular disease, ischemic heart disease, stroke, respiratory disease, and lung cancer. In models that included both PM_2.5_ and NO_2_, the PM_2.5_ associations with mortality from all causes were reduced to about half the size of those in the single-pollutant models, and the estimates became insignificant. The increase in NO_2_ and PM_2.5_ concentration on cardiovascular disease mortality revealed the highest risk among the analyzed results. PM_2.5_ exposition caused a significantly higher risk of death due to congenital heart diseases and ischemic heart diseases, but the effects of PM_2.5_ were attenuated with NO_2_. PM_2.5_ had elevated but insignificant risk estimates for respiratory deaths, whereas neither of the other pollutants was associated with respiratory mortality. For lung cancer, NO_2_ consistently elevated risks in two-pollutant models.

The influence of the increase in NO_2_ and PM_2.5_ concentration on non-accidental mortality was featured as 1–3% of health risk increase [110,122]. The adjustment for PM_2.5_ in NO_2_ increase pointing towards small synergy or no influence was observed, while adjustment for NO_2_ in PM_2.5_ increase revealed a small antagonistic effect. Cesaroni et al. [110] analyzed cause-specific mortality of adults in Rome in accordance with two GIS indicators of traffic exposure (distance to heavy traffic roads with >10,000 vehicles per day, and traffic intensity in 150 m); however, these associations were evaluated separately for NO_2_ and PM_2.5_. In the two-pollutant model, long-term exposure to both NO_2_ and PM_2.5_ were analyzed only in the aspect of non-accidental mortality with HR: 1.02 (95% CI 1.01–1.03) per 10 µg/m^3^ of NO_2_ adjusted for PM_2.5_, and HR: 1.01 (95% CI 0.99–1.02) per 10 µg/m^3^ of PM_2.5_ adjusted for NO_2_.

The single pollutant estimations point to similar non-accidental mortality risk for PM_2.5_ and NO_2_, with HRs of 1.01 (95% CI 1.00–1.02) and 1.01 (95% CI 1.00–1.01), respectively. In the two-pollutant model, the associations were similar to single-pollutant risks; however, PM_2.5_ adjusted for NO_2_ were not statistically significant. Christidis et al. [122] analyzed the data from 1981 to 2016 in a cohort of Canadian communities exposed to low concentrations of PM_2.5_ (average 5.9 µg/m^3^). The unadjusted model had a HR: 0.96 (95% CI 0.92–1.00), which increased to 1.11 (95% CI 1.04–1.18) when adjusted by the socio-economic, behavioral, and contextual covariates. The authors evaluated the impact of individual-level behavioral risk factors on the PM_2.5_ mortality association relationship. The inclusion of behavioral covariates in a model including socioeconomic and ecological covariates lowered the PM_2.5_ hazard ratio by 2% (from 1.13 to 1.11). This modest change in the hazard ratio can indicate that the behavioral covariates were being adequately controlled for by the socio-economic and ecological covariates in the established relationship between PM_2.5_ exposure and non-accidental mortality. In the two-pollutant model, the estimated effect of an increase in NO_2_ on mortality was independent from the PM_2.5_ adjustment, while the HR associated with an increase in PM_2.5_ concentration in the adjustment for NO_2_ was only slightly lower than the HR of a single pollutant (HR: 1.03, 95% CI 1.01–1.05, and HR: 1.03, 95% CI 1.02–1.05 for PM_2.5_ and NO_2_, respectively).

In the article on all-cause mortality by Beelen et al. [76], the authors analyzed the data from 22 European cohorts recruited mainly in the 1990s. Exposure to PM_2.5_ and NO_2_ as separate pollutants resulted in HR: 1.07 (95% CI 1.02–1.13) and 1.01 (95% CI 0.99–1.03), respectively, for a 5 µg/m^3^ increase in PM_2.5_ concentration and 10 µg/m^3^ for NO_2_. However, the health risk of the increase in NO_2_ concentration was not statistically significant (CI includes value 1). In the two-pollutant model, the health effect estimates either for PM_2.5_ adjusted for NO_2_ (HR: 1.06, 95% CI 0.98–1.15) or for NO_2_ adjusted for PM_2.5_ (HR: 1.01, 95% CI 0.97–1.05) did not differ from the single-pollutant model.

The development of chronic kidney disease and confirmed long-term exposure to ambient PM_2.5_ and NO_2_ was examined by Guo et al. [121]. Every 10 μg/m^3^ increase in PM_2.5_ or NO_2_ concentrations was associated with a higher risk of developing chronic kidney disease. The increase in PM_2.5_ concentration resulted in a higher risk of developing chronic kidney disease than the increase in NO_2_ concentration. HR for PM2.5 alone was 1.53 (95% CI 1.07–2.2), while for NO_2_, the risk was eightfold lower (HR: 1.07, 95% CI 1.00–1.15). The adjustment for PM_2.5_ or NO_2_ appeared to be antagonistic in comparison to single-pollutant risk. However, the results were not statistically significant.

In one study [114], the authors analyzed the increase in out-of-hospital cardiac arrest and calculated the ORs per 10 μg/m^3^ increase in NO_2_ levels. Furthermore, they statistically compared the concentrations on control days with the daily average concentrations on the onset day (Lag 0) and on days 1–5 before onset (Lags 1, 2, 3, 4, and 5). In a multi-pollutant model including NO_2_ adjusted for PM_2.5_, the estimated effect was comparable to a single-pollutant risk. For PM_2.5_, OR was 1.07 (95% CI 1.04–1.10), and for NO_2_, it was 1.05 (95% CI 0.98–1.11); however, the calculated OR for PM_2.5_ adjustment was not statistically significant.

Only one study included short-term exposition. Orellano [10], in an extensive review including 196 articles, reported evidence of a positive association between short-term (1 h do 1 day) exposure to air pollutants and all-cause mortality. In the case of single pollutants, the relative risk levels were 1.0065 (95% CI 1.0044–1.0086) and 1.072 (95% CI 1.0059–1.0085) for PM_2.5_ and NO_2_, respectively. As can be seen (Table 3, Table 4 and Table 5), the association values were higher for NO_2_ adjustment in comparison to a single-pollutant model in the case of respiratory diseases and were lower in the case of all-cause mortality; however, in the later, the statistical significance was irrespective.

## 3. Conclusions

A literature review of the health effects of the co-interactions of the pollutants under consideration (PM_2.5_ and NO_2_) showed that it depended on the type of pollutant, the cause of death/disease, and the magnitude of the increase in its concentration.

The increased risk of non-accidental mortality was observed with higher NO_2_ and PM_2.5_ concentrations. NO_2_ was significantly associated with mortality when adjusted for PM_2.5_; however, the estimated effect of PM_2.5_ was no longer significant. With a small increase in concentration, the risk for all-cause mortality was raised from 1 to 3%. NO_2_ exposure adjusted for PM_2.5_ at a high concentration of ambient PM_2.5_ was particularly influential, revealing significant synergy between both pollutants. PM_2.5_ was associated more with cardiovascular disease mortality, whereas NO_2_ exposure was associated more with respiratory mortality. Regardless of which pollutant was major or adjusted, in cardiovascular mortality, the adjustment resulted in a rather antagonistic effect, while in respiratory disease mortality, particularly by pneumonia and lung cancer, NO_2_ exposure adjusted for PM_2.5_ revealed a synergistic effect, and conversely, PM_2.5_ exposure adjusted for NO_2_ brought about an antagonistic effect.

The high correlations between pollutants were associated with serious limitations in the use of multi-pollutant models. Because of this, the health effects of many pollutants are still not fully known. To better understand the health effects of many pollutants, more research needs to be conducted on biological mechanisms, preferably grouping pollutants according to their mode of action.

## Figures and Tables

**Table 1 ijerph-19-14079-t001:** EU limit values and WHO Air Quality Guidelines (AQG) for considered air pollutants.

Air Pollutant	EU Limit Values [21]	WHO AQG [17]	
	Annual ^a^	Annual	Daily
PM_2.5_, µg/m^3^	25	5	15
NO_2_, µg/m^3^	40	10	25

n.a.—not applicable; ^a^ daily values are not available.

**Table 2 ijerph-19-14079-t002:** Risk ratio (RR) vs. odds ratio (OR) vs. hazard ratio (HR) [36].

Parameter	RR	OR	HR
Aim	Determination of relationships in risk potential based on some variable.	Identify the relationship between two variables.	Determine how one group changes relative to the other.
Usage	Informs how an intervention changes risk.	Informs whether there is a relationship between the intervention and the risk; estimates how this relationship occurs.	Informs how intervention changes the rate at which an event is experienced.
Limitations	Only applicable if the study design is representative of the population. Cannot be used for case–control studies.	It can generally be used anywhere but is not always a useful statistic on its own. It exaggerates the risks.	In order to be typically useful, the rate of change within the two groups should be relatively consistent.
Timeline	Static. Summarizes an overall study.	Static. Summarizes an overall study.	Dynamic. Provides information about the way a study progresses over time.
Correct	Most useful and often preferred statistic. Most intuitive and easiest to understand.	Can be more widely used. Can sometimes approximate relative risk in rare cases. Does not require a random sampling. Useful for logistic regression and case–control studies.	Allows a member of association in outcomes in survivorship curves.
Incorrect	Must assume causal direction (change in an independent variable changes the outcome variable). Only effective and useful with randomized something.	Less intuitive and can seem to exaggerate data. Only shows correlation, not causation.	Requires proportional hazards assumption (all data groups must show a roughly linear relationship between number of events and time). Should be reported with median time-to-event.

**Table 3 ijerph-19-14079-t003:** Risk ratio (RR), odds ratio (OR), and hazard ratio (HR), and health effects linked to PM_2.5_ and NO_2_ exposure.

Health Outcomes	Exposure Duration	OR/HR/RR	Mean (Min.-Max.) Concentration µg/m^3^		References
			PM_2.5_	NO_2_	
Non-Accidental Cause Mortality					
Non-accidental causes	Chronic	HR: 1.01 (95% CI 0.99–1.02) 10 µg/m^3^ PM_2.5_ adjusted for NO_2_ HR: 1.02 (95% CI 1.01–1.03) 10 µg/m^3^ NO_2_ adjusted for PM_2.5_	23.0 (7.2–32.1)	43.6 (13.0–75.2)	[110]
	Chronic	HR: 1.02 (95% CI 1.00–1.04) 2.8 µg/m^3^ PM_2.5_ (IQR) adjusted for NO_2_ HR: 1.03 (95% CI 1.01–1.05) 12.5 µg/m^3^ NO_2_ (IQR) adjusted for PM_2.5_	5.9 (0.4–17.2)	8.6 (0.0–69.1)	[122]
Natural cause	Chronic	HR: 1.06 (95% CI 0.98–1.15) 5 µg/m^3^ PM_2.5_ adjusted for NO_2_ HR: 1.01 (95% CI 0.97–1.05) 10 µg/m^3^ NO_2_ adjusted for PM_2.5_	6.6–31	5.2–59.8	[76]
All-cause mortality	Chronic	RR: 1.015 (95% CI 0.980–1.050) 5.03 µg/m^3^ PM_2.5_ (IQR) adjusted for NO_2_ RR: 1.015 (95% CI 0.098–1.050) 7.74 µg/m^3^ NO_2_ (IQR) adjusted for PM_2.5_	14.1 (4.3–25.1)	12.3 (3.0–21.9)	[123]
	Chronic	OR: 1.023 (95% CI 0.814–1.279) 35.6 µg/m^3^ PM_2.5_ (IQR) adjusted for NO_2_ OR: 1.457 (95% CI 1.076–2.152) 16.9 µg/m^3^ NO_2_ (IQR) adjusted for PM_2.5_	80.7 (37.0–142.4)	49.7 (33.6–68.1)	[16]
	Short term (1 h-days)	RR: 1.0004 (95% CI 0.9926–1.0082) 10 µg/m^3^ PM_2.5_ adjusted for NO_2_	5.7–176.7	18.4–99.2 (24-h average) 40.0–161.2 (1 h max.)	[10]
Cardiovascular Disease Mortality					
Cardiovascular	Chronic	RR: 1.043 (95% CI 0.989–1.101) 5.03 µg/m^3^ PM_2.5_ (IQR) adjusted for NO_2_ RR: 1.030 (95% CI 0.987–1.075) 7.74 µg/m^3^ NO_2_ (IQR) adjusted for PM_2.5_	14.1 (4.3–25.1)	12.3 (3.0–21.9)	[123]
	Short term (1 h-days)	RR: 1.0092 (95% CI 0.9945–1.0241) 10 µg/m^3^ PM_2.5_ adjusted for NO_2_	5.7–176.7	18.4–99.2 (24-h average) 40.0–161.2 (1 h max.)	[10]
Congenital heart diseases	Chronic	OR: 1.267 (95% CI 0.643–2.404) 35.6 µg/m^3^ PM_2.5_ (IQR) adjusted for NO_2_ OR: 1.667 (95% CI 1.011–2.738) 16.9 µg/m^3^ NO_2_ (IQR) adjusted for PM_2.5_	80.7 (37.0–142.4)	49.7 (33.6–68.1)	[16]
Ischemic heart disease	Chronic	RR: 1.090 (95% CI 1.015–1.170) 5.03 µg/m^3^ PM_2.5_ (IQR) adjusted for NO_2_ RR: 1.029 (95% CI 0.972–1.090) 7.74 µg/m^3^ NO_2_ (IQR) adjusted for PM_2.5_	14.1 (4.3–25.1)	12.3 (3.0–21.9)	[123]
Out-of-hospital cardiac arrest	Chronic	OR: 1.07 (95% CI 1.03–1.11) 10 µg/m^3^ PM_2.5_ adjusted for NO_2_	76.0 (5.0–476.0)	51.7 (7.8–136.2)	[114]
Respiratory Disease Mortality					
Respiratory	Short term (1h-days)	RR: 1.0135 (95% CI 1.0008–1.0263) 10 µg/m^3^ PM_2.5_ adjusted for NO_2_	5.7–176.7	18.4–99.2 (24-h average) 40.0–161.2 (1 h max.)	[10]
	Chronic	RR: 1.064 (95% CI 0.954–1.185) 5.03 µg/m^3^ PM_2.5_ (IQR) adjusted for NO_2_ RR: 0.973 (95% CI 0.891–1.063) 7.74 µg/m^3^ NO_2_ (IQR) adjusted for PM_2.5_	14.1 (4.3–25.1)	12.3 (3.0–21.9)	[123]
Pneumonia-related	Chronic	OR: 0.961 (95% CI 0.754–1.145) 35.6 µg/m^3^ PM_2.5_ (IQR) adjusted for NO_2_ OR: 1.781 (95% CI 1.011–2.738) 16.9 µg/m^3^ NO_2_ (IQR) adjusted for PM_2.5_	80.7 (37.0–142.4)	49.7 (33.6–68.1)	[16]
Lung cancer	Chronic	RR: 0.985 (95% CI 0.867–1.119) 5.03 µg/m^3^ PM_2.5_ (IQR) adjusted for NO_2_ RR: 1.118 (95% CI 1.010–1.236) 7.74 µg/m^3^ NO_2_ (IQR) adjusted for PM_2.5_	14.1 (4.3–25.1)	12.3 (3.0–21.9)	[123]
Cerebrovascular Disease Mortality					
Stroke	Chronic	RR: 1.019 (95% CI 0.934–1.112) 5.03 µg/m^3^ PM_2.5_ (IQR) adjusted for NO_2_ RR: 1.070 (95% CI 0.998–1.147) 7.74 µg/m^3^ NO_2_ (IQR) adjusted for PM_2.5_	14.1 (4.3–25.1)	12.3 (3.0–21.9)	[123]
Chronic Diseases					
Chronic kidney diseases	Chronic	HR: 1.43 (95% CI 0.98–2.09) 10 µg/m^3^ PM_2.5_ adjusted for NO_2_ HR: 1.05 (95% CI 0.97–1.14) 10 µg/m^3^ NO_2_ adjusted for PM_2.5_	26.6	44.8	[121]

IQR—interquartile range.

**Table 4 ijerph-19-14079-t004:** Risk ratio (RR), odds ratio (OR), and hazard ratio (HR) with corresponding health effects linked to NO_2_ exposure alone and adjusted for PM_2.5_.

Health Outcomes	Exposure Duration	NO_2_ Adjusted for PM_2.5_			NO_2_			Synergistic (+) Antagonistic (−) No Significant Difference (0)		Reference
		HR	OR	RR	HR	OR	RR			
Non-Accidental Cause Mortality										
Non-accidental causes	Chronic	**1.02**			**1.01**				+	[110]
	Chronic	**1.03**			**1.03**				0	[122]
Natural cause	Chronic	1.01			1.01				0	[76]
All-cause mortality	Chronic			1.025			**1.031**		**−**	[123]
	Chronic		**1.457**			**1.383**			+	[16]
Cardiovascular Disease Mortality										
Cardiovascular	Chronic			1.03			**1.048**		**−**	[123]
Congenital heart diseases	Chronic		**1.667**			**2.103**			**− −**	[16]
Ischemic heart disease	Chronic			1.029			**1.066**		**−**	[123]
Respiratory Disease Mortality										
Respiratory	Chronic			0.973			0.999		**−**	[123]
Pneumonia-related	Chronic		**1.781**			**1.74**			+	[16]
Lung cancer	Chronic			1.118			1.111		+	[123]
Cerebrovascular Disease Mortality										
Stroke	Chronic			1.07			**1.078**		**−**	[123]
Chronic Diseases										
Chronic kidney diseases	Chronic	1.05			**1.07**				**−**	[121]

*p* > 0.05 no significant association. Statistically significant risks values are in bold.

**Table 5 ijerph-19-14079-t005:** Risk ratio (RR), odds ratio (OR), and hazard ratio (HR) with corresponding health effects linked to PM_2.5_ exposure alone and adjusted for NO_2_.

Health Outcomes	Exposure Duration	PM_2.5_ Adjusted for NO_2_			PM_2.5_			Synergistic (+) Antagonistic (−) No Significant Difference (0)	Reference
		HR	OR	RR	HR	OR	RR		
Non-Accidental Cause Mortality									
Non-accidental causes	Chronic	1.01			**1.01**			0	[110]
	Chronic	**1.02**			**1.03**			**−**	[122]
Natural cause	Chronic	1.06			**1.07**			**−**	[76]
All-cause mortality	Chronic			1.015			**1.032**	**−**	[123]
	Chronic		1.023			1.155		**−**	[16]
	Short term (1 h-days)			1.00			**1.0065**	**−**	[10]
Cardiovascular Disease Mortality									
Cardiovascular	Chronic			1.043			**1.064**	**−**	[123]
	Short term (1 h-days)			1.009			**1.0092**	0	[10]
Congenital heart diseases	Chronic		1.267			**1.653**		**− −**	[16]
Ischemic heart disease	Chronic			**1.09**			**1.111**	**−**	[123]
Out-of-hospital cardiac arrest	Chronic		**1.07**			1.07		0	[114]
Respiratory Disease Mortality									
Respiratory	Short term (1 h-days)			**1.014**			**1.0073**	+	[10]
	Chronic			1.064			**1.046**	+	[123]
Pneumonia-related	Chronic		0.961			1.171		**− −**	[16]
Lung cancer	Chronic			0.985			1.062	**−**	[123]
Cerebrovascular Disease Mortality									
Stroke	Chronic			1.019			1.065	**−**	[123]
Chronic Diseases									
Chronic kidney diseases	Chronic	1.43			**1.53**			**− −**	[121]

*p* > 0.05 non-significant association. Statistically significant risks values are in bold.

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
