# Peer review of "Synergistic or Antagonistic Health Effects of Long- and Short-Term Exposure to Ambient NO2 and PM2.5: A Review"

_ijerph, 2022, doi:10.3390/ijerph192114079_

Round 1

Reviewer 1 Report

This manuscript explain the effects of PM2.5 and NO2 and the combined pollutants toward human health. Generally, this review paper  lacked of critical discussion. The authors summarized the research findings from previous literature and presented in this manuscript especially on the effects of individual and combined air pollutants. Some of the information specifically on the effects of air pollution toward human health was too general and well known. Listed below are the comments for this manuscript:

1. Title - I suggest that "review" should be included in the title.

2. The abstract was too general. Need to include specific outcomes from the review and the sgnificance of this study. No word of synergistic or antagonistic were given in the abstract.

3. Since this is a review paper, I think that no methodology is needed for this paper. 

4. Results & Discussion - in this section, the authors should critically review the effects of individual and combined pollutants according to the type such as synergistic/ antagonistic/ inhibition etc. Your title did not represent the body of the manuscript. The authors only summarized previous research finding in this section.

5. Conclusion - The conclusion was too general.

The detail comments were given in the attached document.

Author Response

October 17th, 2022

Manuscript ID: IJERPH (ISSN 1660-4601)

Title: " Synergistic or antagonistic health effects of long- and short-term exposure to ambient NO2 and PM2.5"

Response to the comments of Reviewer 1

We appreciate your response and the helpful suggestions for improving the presentation of our study. Below please find details on all modifications as they relate to your comments on the original manuscript (marked in red).

Reviewer’s remarks:

This manuscript explain the effects of PM2.5 and NO2 and the combined pollutants toward human health. Generally, this review paper lacked of critical discussion. The authors summarized the research findings from previous literature and presented in this manuscript especially on the effects of individual and combined air pollutants. Some of the information specifically on the effects of air pollution toward human health was too general and well known. Listed below are the comments for this manuscript:

  1. Title - I suggest that "review" should be included in the title.

Corrected

  1. The abstract was too general. Need to include specific outcomes from the review and the sgnificance of this study. No word of synergistic or antagonistic were given in the abstract.

The abstract has been corrected, as presented below:

NO2 and PM2.5 have been selected as both originate from combustion processes and are considered to be the main pollutants associated with traffic, moreover both elicit oxidative stress responses. An answer to question if synergistic or antagonistic health effects of combined pollutants are demonstrated by pollutants monitored in ambient air is not explicit. Among the analyzed studies only a few revealed statistical significance. Exposure to a single pollutant (PM2.5 or NO2) was mostly associated with a small increase in non-accidental mortality (1.01-1.03). PM2.5 increase of <10 µg/m3 adjusted for NO2 as well as NO2 adjusted for PM2.5 resulted in a bit lower health risk than a single pollutant. In the case of cardiovascular heart disease mortality evoked by exposure to PM2.5 or NO2 adjusted for NO2 and PM2.5, respectively revealed an antagonistic effect on health risk compared to the single pollutant. Both short- and long-term exposure to PM2.5 or NO2 adjusted for NO2 and PM2.5, respectively revealed a synergistic effect appearing as higher mortality from respiratory diseases.

  1. Since this is a review paper, I think that no methodology is needed for this paper.

The heading “methodology” has been deleted, so the numeration of headings has changed.

  1. Results & Discussion - in this section, the authors should critically review the effects of individual and combined pollutants according to the type such as synergistic/ antagonistic/ inhibition etc. Your title did not represent the body of the manuscript. The authors only summarized previous research finding in this section.

The title Results & Discussion has been changed into Considerations and discussion. The whole section has been deeply changed (lines 431-537).

  1. Conclusion - The conclusion was too general.

Conclusions have been deeply changed, as presented below:

In the title the authors addressed the question whether by pollutants monitored in ambient air demonstrated synergistic or antagonistic health effects of combined pollutants. The answer depends upon the type of pollutant, cause of mortality and increase of pollutant’s concentration.

The increased risk of non-accidental mortality was observed with higher NO2 and PM2.5 concentrations. NO2 was significantly associated with mortality when adjusted for PM2.5, however the estimated effect of PM2.5 was no longer significant. With small increase of concentration the risk for all-cause mortality raised from 1-3%. NO2 exposure adjusted for PM2.5 at high concentration of ambient PM2.5 was particularly influential revealing significant synergy between both pollutants. PM2.5 was more associated with cardiovascular disease mortality, whereas NO2 exposure with respiratory mortality. Regardless which pollutant was major or adjusted, in cardiovascular mortality the adjustment resulted in rather antagonistic effect, while in respiratory diseases mortality, particularly by pneumonia and lung cancer, NO2 exposure adjusted for PM2.5 revealed synergistic effect, while PM2.5 exposure adjusted for NO2 brought antagonistic effect.

The high correlations between pollutants are associated with serious limitations in the use of multi-pollutant models. Because of this, the health effects of many pollutants are still not fully known. To better understand the health effects of many pollutants, more research needs to be done on biological mechanisms, preferably grouping pollutants according to their mode of action.

Since the Reviewer in the additional comments at the manuscript advised to delete Table 3 has been removed from the text. Additionally, as suggested by the reviewer, subscripts were introduced at PM2.5.

Again, we would like to express our appreciation for your efforts and helpful comments. Enclosed, please find the revised version of the paper.

Yours sincerely,

Anna Mainka and Magdalena Żak

Reviewer 2 Report

- authors should discuss the role of dietary habits in this context. They can take into account the paper from Scicchitano P et al. Journal of Functional Foods 2014;6:11-32

- please include a representative figure for this paper in order to improve ther readability of the text.

Author Response

October 17th, 2022

Manuscript ID: IJERPH (ISSN 1660-4601)

Title: " Synergistic or antagonistic health effects of long- and short-term exposure to ambient NO2 and PM2.5"

Response to the comments of Reviewer 2

We appreciate your response and the helpful suggestions for improving the presentation of our study. Below please find details on all modifications as they relate to your comments on the original manuscript (marked in blue).

Reviewer’s remarks:

- authors should discuss the role of dietary habits in this context. They can take into account the paper from Scicchitano P et al. Journal of Functional Foods 2014;6:11-32

The reference has been added in lines 114-115.

- please include a representative figure for this paper in order to improve the readability of the text.

Graphical abstract has been added.

Again, we would like to express our appreciation for your efforts and helpful comments. Enclosed, please find the revised version of the paper.

Yours sincerely,

Anna Mainka and Magdalena Żak

Reviewer 3 Report

The following main issues should also be considered before the paper is published.

 As a review, some studies published in 2022 should also be considered, the following is the recent related literature browsed by the reviewers, you may supplement these but are not limited to the below.

 Line 25-26, https://doi.org/10.3390/ijerph19138078, Short-Term Effects of PM10, NO2, SO2, and O3 on Cardio-Respiratory Mortality

Section: Short-term exposure to PM2.5, https://doi.org/10.1016/j.scs.2022.104055

Section: Short-term exposure to NO2, https://doi.org/10.3390/ijerph19010495

Section: NO2 health effect mechanismhttps://doi.org/10.1016/j.toxrep.2022.03.045

Line221, https://doi.org/10.3390/rs14030599, prediction of the PM2.5 concentration.

Line 257-258, https://doi.org/10.1016/j.envint.2022.107331, it provides data of PM2.5-related mortality burdens for 296 Chinese cities during 2015–2019

 Line72, this part is recommended to be indented or numbered.

Line239, 3.1.1 is recommended to add.

 Line3833.1 should be 3.3

 Line413-466, this section was not organized well, it seems that six paragraphs just simply list the context of six pieces of literature, it is recommended to rewrite this part.

Author Response

October 17th, 2022

Manuscript ID: IJERPH (ISSN 1660-4601)

Title: " Synergistic or antagonistic health effects of long- and short-term exposure to ambient NO2 and PM2.5"

Response to the comments of Reviewer 3

We appreciate your response and the helpful suggestions for improving the presentation of our study. Below please find details on all modifications as they relate to your comments on the original manuscript (marked in green).

Reviewer’s remarks

The following main issues should also be considered before the paper is published.

As a review, some studies published in 2022 should also be considered, the following is the recent related literature browsed by the reviewers, you may supplement these but are not limited to the below.

Line 25-26, https://doi.org/10.3390/ijerph19138078, Short-Term Effects of PM10, NO2, SO2, and O3 on Cardio-Respiratory Mortality

The reference of Adebayo-Ojo et al. as well as additional references have been added (lines 34-34).

Section: Short-term exposure to PM2.5, https://doi.org/10.1016/j.scs.2022.104055

In our opinion the reference of Gao et al. will better fit to the introduction (lines 102-106):

To assess the multi-index analysis of ecological risks from the source perspective a Geographic Information System (GIS) and Remote Sensing (RS) can be used. Among indicators reflecting environmental risks, the pressure from urban expansion, land use, and degradation as well as cropland proportion can be included [24]. 

Section: Short-term exposure to NO2, https://doi.org/10.3390/ijerph19010495

The following text has been added (lines 342-345):

Adebayo-Ojo et al.[4] point to a significant increase of hospital admissions for respiratory disease per lower increase of NO2 concentration (interquartile range IQR of 7.3 µg/m3). A positive association among all ages was 2.3; however, among children below 15 years old, the estimate increased to 3.1%.

Section: NO2 health effect mechanism,https://doi.org/10.1016/j.toxrep.2022.03.045

The reference has been added (line 379).

And the following text from the reference has been added in section PM2.5 and NO2 (lines 409-411):

Based on 7 years of air pollutants measurements in Iran, analyzed by AirQ+ model, Naghan et al. [106] observed that the short-term effects of PM2.5 on health were greater than those of NO2. However, the long-term effects of NO2 were greater than PM2.5.

Line221, https://doi.org/10.3390/rs14030599, prediction of the PM2.5 concentration.

The following text from the reference has been added in section PM2.5 and NO2 (lines 227-230):

Unfortunately, the number of air monitoring ground stations is limited, and the spatial distribution is discontinuous, so to obtain a fine-grained spatiotemporal distribution of PM2.5 a retrieval model can be used [57].

Line 257-258, https://doi.org/10.1016/j.envint.2022.107331, it provides data of PM2.5-related mortality burdens for 296 Chinese cities during 2015–2019

The following text from the reference has been added in section PM2.5 and NO2 (lines 269-271):

Han et al. [74] based on the measurements, conducted from 2015 to 2019 in 296 cities across China, estimated that long-term exposure to PM2.5 levels exceeding current WHO guidelines (5 µg/m3) was associated with 17% average all-cause mortality.

Line72, this part is recommended to be indented or numbered.

Corrected

Line239, 3.1.1 is recommended to add.

Corrected

Line383,3.1 should be 3.3

Corrected

Line413-466, this section was not organized well, it seems that six paragraphs just simply list the context of six pieces of literature, it is recommended to rewrite this part.

The title Results & Discussion has been changed into Considerations and discussion. The whole section has been deeply changed lines: 431-537

Again, we would like to express our appreciation for your efforts and helpful comments. Enclosed, please find the revised version of the paper.

Yours sincerely,

Anna Mainka and Magdalena Żak

Round 2

Reviewer 1 Report

Overall, all the issues addressed in the first revision had been addressed well. The manuscript are good to be published. 

Author Response

Dear Reviewer,

We appreciate your response and the helpful suggestions for improving the presentation of our review.

Yours sincerely,

Anna Mainka and Magdalena Żak

Reviewer 3 Report

After careful revision, I think the current version is suitable for publication in the journal IJERPH.

Author Response

(The authors gave the same response as above.)
